# Scalable and Configurable Electrical Interface Board for Bus System Development of Different CubeSat Platforms

**Marloun Sejera** [1,2,*] **, Takashi Yamauchi** [1] **, Necmi Cihan Orger** [1] **, Yukihisa Otani** [1] **and Mengu Cho** [1]

1 Laboratory of Lean Satellite Enterprises and In-Orbit Experiments (LaSEINE), Department of Electrical and Space Systems Engineering, Kyushu Institute of Technology, Kitakyushu 804-8550, Japan
2 School of Electrical, Electronics, and Computer Engineering, Mapua University, Manila 1002, Philippines
* Correspondence: sejera.marloun-pelayo148@mail.kyutech.jp or mpsejera@mapua.edu.ph

**Abstract:** A flight-proven electrical bus system for the 1U CubeSat platform was designed in the BIRDS satellite program at the Kyushu Institute of Technology. The bus utilizes a backplane board as the mechanical and electrical interface between the subsystems and the payloads. The electrical routes on the backplane are configured by software using a complex programmable logic device (CPLD). It allows for reusability in multiple CubeSat projects while lowering costs and development time; as a result, resources can be directed toward developing the mission payloads. Lastly, it provides more time for integration and system-level verification, which are critical for a reliable and successful mission. The current trend of CubeSat launches is focused on 3U and 6U platforms due to their capability to accommodate multiple and complex payloads. Hence, a demonstration of the electrical bus system to adapt to larger platforms is necessary. This study demonstrates the configurable electrical interface board's scalability in two cases: the capability to accommodate (1) multiple missions and (2) complex payload requirements. In the first case, a 3U-size configurable backplane prototype was designed to handle 13 mission payloads. Four CPLDs were used to manage the limited number of digital interfaces between the existing bus system and the mission payloads. The measured transmission delay was up to 20 ns, which is acceptable for simple serial communications such as UART and SPI. Furthermore, the measured energy consumption of the backplane per ISS orbit was only 28 mWh. Lastly, the designed backplane was proven to be highly reliable as no bit errors were detected throughout the functionality tests. In the second case, a configurable backplane was implemented in a 6U CubeSat with complex payload requirements compared to the 1U CubeSat platform. The CubeSat was deployed in ISS orbit, and the initial on-orbit results indicated that the designed backplane supported missions without issues.

**Keywords:** CubeSat; electrical interface; scalability; bus system development

## 1. Introduction

A CubeSat is a class of satellites with a defined size and form factor. A 1U CubeSat, for example, has dimensions of $10 \times 10 \times 11.35$ cm$^3$ and a mass of up to 2 kg, as defined in the CubeSat Design Specifications from California Polytechnic State University (CalPoly) [1]. The document describes the mechanical, electrical, and operational specifications of Cube-Sats from 1U to 12U. However, it does not cover how the components in a CubeSat, i.e., both the bus and the payload, are interfaced. This lack of such a definition allows CubeSat developers the freedom to choose which interface method to use. More importantly, this aspect could cause incompatibility issues between components, and solving these issues could considerably consume time that could be used for other verification activities to ensure mission success [2]. Furthermore, CubeSat projects can be developed by multiple collaborators, and clearly defining an interface standard between developers during the initial phase could prevent project delays, compatibility issues, and increased costs while improving overall mission success.

The first CubeSats developed used the stacking approach, where components were placed on top of each other using a connector. This interfacing method follows the PC/104 specifications [3] employed in embedded computers. The adoption of this specification to CubeSats defines the wiring harness, the printed circuit board (PCB) footprint, and the mechanical mounting of the boards, while the boards are stacked using 104-pin connectors. One of the first developers to employ the PC/104 specification in CubeSat applications is Pumpkin, Inc., who introduced the CubeSat Kit Bus (CSKB) [4]. The CSKB has become the de facto standard in CubeSat design and has been adopted by many commercial CubeSat developers.

However, using the PC/104 specification in CubeSats has several issues. According to a survey in [5], 51% of 36 respondents agreed that the size of the connector is too big. Ref. [6] confirmed that the connector occupies up to 20% of the PCB space. This limits the board designers in placing components on the board, resulting in low PCB utilization. The stacking height of the connector is also considerable, making the spacing between boards particularly wide. Another issue is that the number of pins interconnecting the board is often not utilized to its fullest extent. This may increase the risk of human error when assigning pins and mapping during the development and integration phases. Even though there could be many unused pins, a harness is extensively used in stacked systems. Lastly, top boards need to be disassembled if there is a need to take out a middle board, especially during troubleshooting. This leads to additional development and integration time.

There have been multiple efforts to resolve issues in using PC/104, specifically in terms of its connector. ISIS, for example, started to consider alternatives to CSKB connectors [7]. The company introduced CSKB Lite, two 28-pin connectors with just enough pins for full utilization. It is also backward-compatible with the standard CSKB connector. In the case of Nagoya University's NUcube satellites, where high-density interfaces are necessary [8], 144-pin connectors were used but with only a 9 mm stack-up height to reduce the volume occupied by the PCBs and a pitch of 0.50 mm to allow more space for components on the boards. This, in turn, made the connector incompatible with CSKB. Lastly, Korea Space University's KAUSAT-5 CubeSat used a flexible flat cable (FFC) connector instead of PC/104 to save volume and mass [6]. Despite these efforts, issues such as extensive use of harnesses and difficulty when assembling and disassembling were not addressed.

Another interface method uses a dedicated PCB called a backplane board that provides mechanical and electrical connections between the bus and the payloads. UWE-3, a 1U CubeSat from the University of Würzburg in Germany, first carried out the use of a backplane and became the reference for UNISEC Europe's CubeSat specification interface document (CSID) interface [9,10]. The BIRDS satellite program at the Kyushu Institute of Technology (Kyutech) in Japan also adopted the backplane board approach to its 1U CubeSats as an interface to the bus and the payload [11]. The bus and payload boards have 50-pin male connectors connected to the 50-pin female connectors of the backplane board. These connectors have a smaller form factor, which provides more space for electronic components on the bus and payload boards. The spacing between boards can also be adjusted by moving the female connectors on the backplane board. This provides efficient utilization of the limited volume of a CubeSat. In addition, power lines, as well as analog and digital signals, are routed through the PCB. This greatly reduces the use of the wiring harness, which is considered one of the fundamental reasons for satellite failure. Lastly, the backplane approach makes satellite assembly and disassembly significantly more straightforward. Therefore, it allows integration and troubleshooting to be performed in a shorter time.

A survey was conducted to determine the interface method used in CubeSats launched from 2003 to 2019 [12]. Of the 397 CubeSats surveyed, only 170 CubeSats had an identified interface since the sources of information (e.g., web pages, papers, conference papers, etc.) did not provide the details. A total of 137 CubeSats used the PC/104 interface, which is about 80% of the total identified CubeSats. However, the use of backplanes on CubeSats started gaining favor in 2013, with 24 satellites launched from then until 2019. For example,

CalPoly's Aerospace Engineering Department developed its own kit as an educational platform for satellite development known as CalPoly CubeSat Kit MK1. Its internal configuration uses a backplane that can connect five boards through its 48-pin female connectors [13]. However, this backplane's design lacks flexibility because of changes in interface definition from one CubeSat project to the other. These changes lead to the complete reproduction of the backplane, adding cost and development time [14].

The third generation of the BIRDS project introduced a standard software-configurable backplane board as one of its technology demonstrations [14]. A complex programmable logic device (CPLD) was placed on the backplane, and digital signals between the bus and the payload were routed through the CPLD. By reprogramming the chip, rerouting can be performed without changes in the hardware design. This makes the backplane flexible and reusable by future satellites with minimal modification while saving time and cost. The configurable backplane was proven to work in space during satellite deployment at the International Space Station (ISS) in June 2019 until it was deorbited in October 2021. The software-configurable backplane has become an integral part of the BIRDS 1U bus architecture. The standard bus, however, has limited digital interfaces that constrain the number of payloads a CubeSat can carry out.

The demand for launching CubeSats is on a continuous uptrend, and most CubeSats being launched are 3U and 6U platforms mainly because they can accommodate multiple and complex payloads. Implementing a configurable interface board on larger CubeSat platforms could provide benefits similar to those in the 1U platform, such as a reduction in development throughput and cost.

This paper aims to demonstrate how the configurable interface board can be scaled up and adapted to different CubeSat sizes using the BIRDS 1U standard bus system. The novelty of the present work is described as a confirmation of the configurable interface board's flexibility in absorbing challenges encountered when scaling up to larger CubeSat platforms. Since implementing a configurable backplane to larger platforms such as 3U or 6U CubeSats has not been achieved before, several challenges such as managing communication between the existing standard bus system and multiple mission payloads, as well as meeting the mission requirements of complex payloads, are extensively covered in this study. In addition, the contribution of this paper is that it presents a 3U configurable backplane designed to manage several missions on the limited number of available electrical interfaces of a standard bus. The design concept could benefit satellite developers who provide hosted payload services where bus resources are maximized to accommodate as many payloads as possible. This paper also demonstrates the modifications in a bus system necessary to scale up and handle complex missions in a W6U CubeSat. The CubeSat was deployed from the ISS in March 2022, and it has been successfully supporting the execution of the missions.

This paper is composed of five sections. Section 2 discusses the standard 1U bus system. It also discusses two backplane designs in different CubeSat platforms—a 3U backplane prototype for multiple payloads and a backplane for a W6U CubeSat with complex mission requirements. Section 3 discusses the tests conducted and the results of the two backplane designs. Lastly, Sections 4 and 5 present the Discussion and Conclusion.

## 2. BIRDS 1U Standard Bus System

The BIRDS Program of Kyutech is an educational, capacity-building satellite program that aims to empower participants from non-space-faring countries to lead or start satellite projects in their home countries [15]. In the program, the graduate students from the participating countries gain hands-on experience on how to design, develop, test, and operate CubeSats. A total of thirteen countries have participated in the program, nine of which built the first satellite in their countries. The program has deployed a total of seventeen satellites in its five generations of constellations from 2015 to 2022—five in BIRDS-1 and three each in BIRD-2, BIRDS-3, BIRDS-4, and BIRDS-5. A total of sixteen satellites are 1U CubeSats, while one of the three satellites in BIRDS-5 is a 2U CubeSat.

Kyutech has made the BIRDS 1U bus system available as open-source information [16,17]. CubeSat developers can gain full access to all information necessary to build a 1U satellite, and these include technical drawings, source code, PCB design, assembly and testing procedures, test reports, and interface control documents (ICDs). This effort allows more people to develop their satellites in an easier, faster, and cheaper way. At present, there are two universities, two high school projects, and a company in Japan that are benefiting from this initiative. In addition, BIRDS members from Malaysia, Mongolia, the Philippines, and Sri Lanka are making satellites in their countries using the BIRDS 1U bus.

Figure 1 shows the internal boards of a 1U BIRDS satellite. The components are described below.

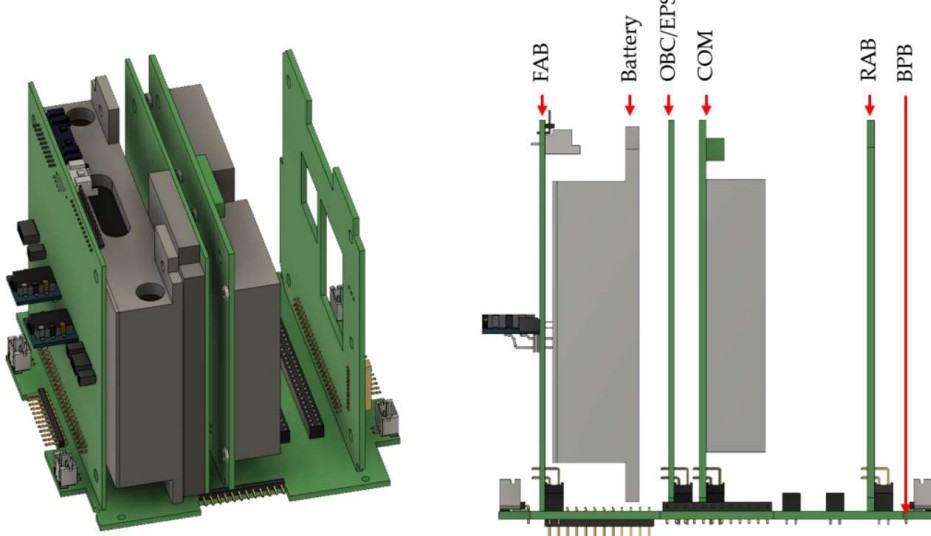

**Figure 1.** BIRDS 1U satellite internal boards.

- *Front access board (FAB)*. This board offers external interfaces or umbilical connections. These interfaces or connectors give users access to subsystem microcontrollers for programming and debugging, monitoring voltage levels, and battery charging. The FAB is also in charge of managing the batteries, collecting the power produced by the solar panels, and controlling electrical power safety. The FAB microcontroller is a Microchip PIC 16F1789 that manages the housekeeping data, including the battery health and solar panel power generation.
- *Battery*. The bus uses six commercial nickel–metal hydride (NiMH) batteries in a three-series, two-parallel (3S2P) configuration. The batteries are housed in a battery box and are connected to the FAB via a 10-pin connector.
- *On-board computer/electrical power system (OBC/EPS)*. The board serves three primary functions. A dedicated Microchip PIC microcontroller handles each process. The main PIC (PIC 18F67J94) oversees command and data handling. The uplink command is received and executed by the main PIC. It also obtains the FAB PIC housekeeping data and mission data for downlink. The COM PIC (PIC 16F1789) manages communication between the radio transceiver and the main PIC. It sends commands from the radio transceiver to the main PIC. It also forwards data from the main PIC to the radio transceiver for downlink. The reset PIC (PIC 16F1789) carries out the electrical power subsystem, and it is in charge of power distribution to other subsystems and the payload. The radio transceiver and antenna deployment system use two unregulated power lines, and there are also two 3.3 V power lines and one 5 V line that the mission payload can use. All three microcontrollers are connected via universal asynchronous receiver/transmitter (UART) serial interfaces to form a ring network.
- *Communication (COM) board*. This board is a dedicated radio transceiver that handles both uplink and downlink communication from and to the ground. The transceiver

employs Gaussian minimum-shift keying (GMSK) modulation with a baud rate of 4800 bps in both directions and an AX.25 protocol for data format. The frequency of operation is in the ultra-high frequency (UHF) amateur band.

- *Rear access board (RAB)*. This board provides external access to the mission payload for programming and debugging.
- *Backplane board (BPB)*. This backplane connects all internal and external boards electrically and mechanically. Figure 2 shows the top side of a backplane board. The internal boards are connected to the backplane via 50-pin female connectors with a 2 mm pitch (C101 to C106). The pins are assigned as power lines, digital lines (for programming, debugging, and data communication), and analog lines. The connector has two pins assigned to each power line and the system ground. The same pins are set to all the connectors. Since the maximum current for each pin is rated as 1 A, the power line current is limited to 2 A. The remaining pins of the connectors are for digital lines and analog lines.

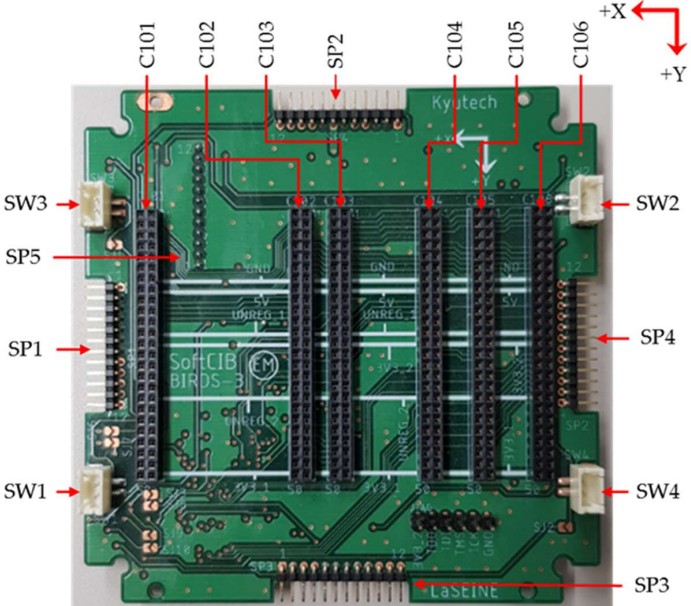

**Figure 2.** Backplane board (top side).

The antenna panel and four solar panels are connected to the backplane via 12-pin male connectors with a 2.54 mm pitch (SP1 to SP5). The fifth solar panel is connected directly to the FAB. The connectors in the solar panels route the generated power and temperature readings to the FAB, and the connector on the antenna panel provides an unregulated power line to the antenna deployment system. In addition, the backplane also includes two-pin male connectors (SW1 to SW4) for the deployment switches connected to the satellite structure.

The space between the COM (C103) and RAB (C104) connectors is allocated for the mission payload. The maximum board thickness of the payload that can be accommodated is 22.35 mm. Up to two mission boards (C104, C105) can be placed in the given space. It is possible to customize the backplane board to reduce the number of 50-pin female connectors to one and shift its position.

A satellite ICD contains the overall information on the mechanical and electrical interfaces between the components. The connector pin assignment is an example of an electrical interface specification found in this document, and the satellite developer can use the pin assignment to determine how components are connected. Table 1 shows the pin assignment of the 50-pin connector on the OBC/EPS board. In addition, it specifies the name of each pin, the connected microcontroller, the pin route, the rated voltage and current, and the protocol and baud rate for digital pins. The pins are named according to

their function or connection. For example, PROG_GIO_1 denotes the first general I/O pin for programming. Another example is OBC-COM_1, which denotes the first pin connecting the OBC/EPS board and COM board. Pins 9–12, 21, 22, 27, 28, 37, 38, and 42 are linked to the main PIC and routed to the mission board. These 11 pins serve as digital interfaces for the bus and the payload, whereas pins 31–34 are serial peripheral interface (SPI) connections to the flash memory (FM) that the main PIC and mission payload share.

**Table 1.** OBC/EPS board pin assignment.

| Pin No. | Pin Name | MCU | Destination | Voltage | Current | Baud Rate | Protocol |
|---|---|---|---|---|---|---|---|
| 1 | PROG_GIO_1 | All | FAB | 3.3 V | <30 mA | - | PIC |
| 2 | PROG_GIO_2 | All | FAB | 3.3 V | <30 mA | - | PIC |
| 3 | No connection | - | - | - | - | - | - |
| 4 | PROG_GIO_4 | Reset | FAB | 3.3 V | <30 mA | - | PIC |
| 5 | PROG_GIO_5 | COM | FAB | 3.3 V | <30 mA | - | PIC |
| 6 | PROG_GIO_6 | Main | FAB | 3.3 V | <30 mA | - | PIC |
| 7 | OBC-COM_1 | COM | COM board | 3.3 V | <30 mA | 115,200 | RS232 |
| 8 | OBC-COM_2 | COM | COM board | 3.3 V | <30 mA | 115,200 | RS232 |
| 9 | FAB_to_RAB_GIO_3 | Main | Mission | 3.3 V | <30 mA | - | DIO |
| 10 | FAB_to_RAB_GIO_4 | Main | Mission board | 3.3 V | <30 mA | - | DIO |
| 11 | FAB_to_RAB_GIO_5 | Main | Mission board | 3.3 V | <30 mA | 115,200 | RS232 |
| 12 | FAB_to_RAB_GIO_6 | Main | Mission board | 3.3 V | <30 mA | 115,200 | RS232 |
| 13 | GND_SYS | - | All boards | GND | 1 A | - | Power |
| 14 | GND_SYS | - | All boards | GND | 1 A | - | Power |
| 15 | SUP_5 V0 | - | All boards | 5 V | 1 A | - | Power |
| 16 | SUP_5 V0 | - | All boards | 5 V | 1 A | - | Power |
| 17 | FAB_to_OBC_GIO_1 | Main | FAB | 3.3 V | <30 mA | 9600 | RS232 |
| 18 | FAB_to_OBC_GIO_2 | Main | FAB | 3.3 V | <30 mA | 9600 | RS232 |
| 19 | FAB_to_OBC_GIO_3 | Main | FAB | 3.3 V | <30 mA | 9600 | RS232 |
| 20 | FAB_to_OBC_GIO_4 | Main | FAB | 3.3 V | <30 mA | 9600 | RS232 |
| 21 | CPLD_8 | Main | Mission board | 3.3 V | <30 mA | 9600 | RS232 |
| 22 | CPLD_9 | Main | Mission board | 3.3 V | <30 mA | 9600 | RS232 |
| 23 | SUP_UNREG_1 | - | All boards | Unreg | 1 A | - | Power |
| 24 | SUP_UNREG_1 | - | All boards | Unreg | 1 A | - | Power |
| 25 | SUP_3 V3_2 | - | All boards | 3.3 V | 1 A | - | Power |
| 26 | SUP_3 V3_2 | - | All boards | 3.3 V | 1 A | - | Power |
| 27 | CPLD_10 | Main | Mission board | 3.3 V | <30 mA | 9600 | RS232 |
| 28 | CPLD_11 | Main | Mission board | 3.3 V | <30 mA | 9600 | RS232 |
| 29 | RAW_POWER | - | FAB | Raw power | 1 A | - | Power |
| 30 | RAW_POWER | - | FAB | Raw power | 1 A | - | Power |
| 31 | CPLD_12 | Memory | Mission board | 3.3 V | <30 mA | 1,000,000 | SPI |
| 32 | CPLD_13 | Memory | Mission board | 3.3 V | <30 mA | 1,000,000 | SPI |
| 33 | CPLD_14 | Memory | Mission board | 3.3 V | <30 mA | 1,000,000 | SPI |
| 34 | CPLD_15 | Memory | Mission board | 3.3 V | <30 mA | 1,000,000 | SPI |
| 35 | SUP_UNREG_2 | - | All boards | Unreg | 1 A | - | Power |
| 36 | SUP_UNREG_2 | - | All boards | Unreg | 1 A | - | Power |
| 37 | CPLD_16 | Main | Mission board | 3.3 V | <30 mA | 9600 | RS232 |
| 38 | CPLD_17 | Main | Mission board | 3.3 V | <30 mA | 9600 | RS232 |
| 39 | Kill_SW | Main | FAB | 3.3 V | <30 mA | - | DIO |
| 40 | No connection | - | - | - | - | - | - |
| 41 | No connection | - | - | - | - | - | - |
| 42 | CPLD_18 | Main | Mission | 3.3 V | <30 mA | - | DIO |
| 43 | OBC-COM_3 | COM | No connection | 3.3 V | <30 mA | - | DIO |
| 44 | OBC-COM_4 | COM | COM board | 3.3 V | <30 mA | - | DIO |
| 45 | OBC-COM_5 | COM | COM board | 3.3 V | <30 mA | - | DIO |
| 46 | OBC-COM_6 | COM | COM board | 3.3 V | <30 mA | - | DIO |
| 47 | OBC-COM_7 | COM | COM board | Analog | <30 mA | - | Analog |
| 48 | OBC-COM_8 | COM | COM board | Analog | <30 mA | - | Analog |
| 49 | SUP_3 V3_1 | - | All boards | 3.3 V | 1 A | - | Power |
| 50 | SUP_3 V3_1 | - | All boards | 3.3 V | 1 A | - | Power |

Figure 3 shows the bottom side of the backplane, where all active devices are placed. This is to avoid interference with the components at the top. One of the active components is a CPLD that can be programmed to perform specific logical functions. The CPLD is a lattice semiconductor ispMACH LC4256ZE, an ultralow power device that uses 1.8 V LVCMOS (low-voltage CMOS) technology with a standby current of 13 uA. A voltage regulator that uses 3.3 V input from one of the power lines supplies 1.8 V to the integrated circuit. In addition, a joint test action group (JTAG) connection is used to program the chip.

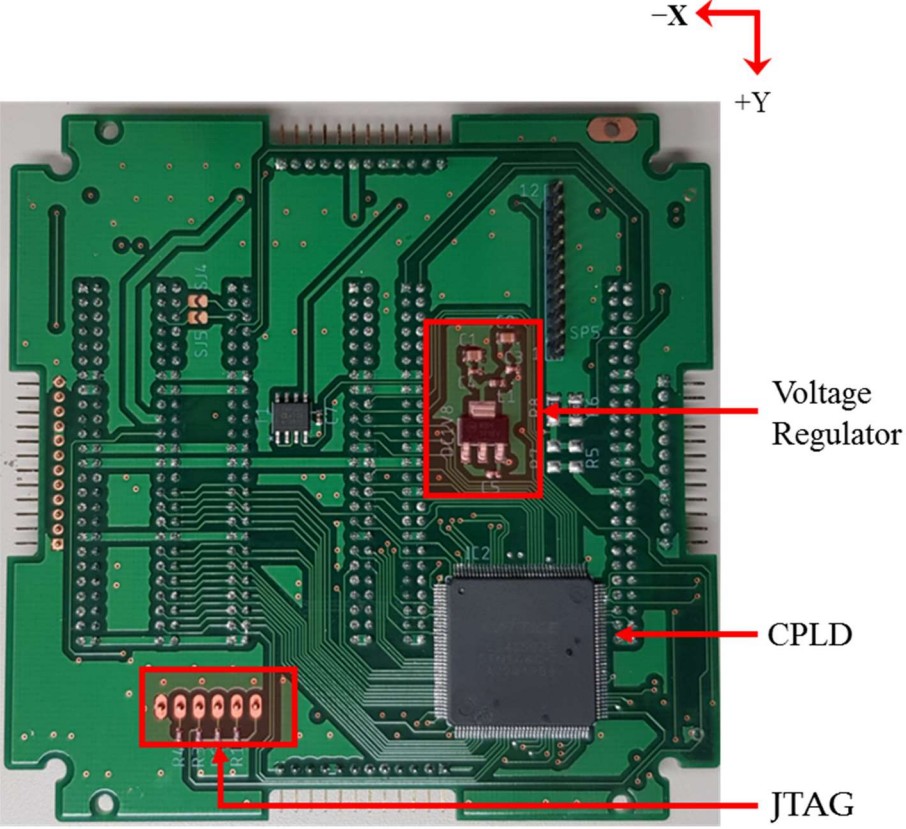

**Figure 3.** Backplane board (**bottom** side).

Table 2 represents the list of CubeSats with their CPLD families and corresponding functionalities. The CPLDs are most commonly implemented within the telemetry command and data-handling subsystem. On the other hand, the CPLD in BIRDS-3 was implemented on the backplane board as an interface between the PIC and the mission payloads. The table also shows the propagation delay and internal voltage supply of the CPLD according to the datasheet. The propagation delay varies depending on the chip package, and the data are based on the 144-pin quad-flat package (QFP) chip if not indicated by the resources. The ispMach4000 ZE CPLD was chosen for its optimal performance in terms of propagation delay and power consumption in addition to ease of availability, implementation, and cost.

Figure 4 illustrates how data communication is performed between the bus system and mission boards. The 11 digital interfaces from the bus are routed to the mission boards through the CPLD. Depending on the implementation, these interfaces can be UART, SPI, digital input/output (DIO), or their combination. There is also an SPI line for transferring mission data to the shared FM in the OBC/EPS board, and the CPLD manages these digital interfaces to allow mission payload communication to the bus system. In addition, Figure 5 shows how a CPLD on the backplane operates where the CPLD is programmed as a voltage follower, with the output pin logic levels matching the paired input pin. Digital interfaces can be rerouted without requiring hardware changes by reprogramming the CPLD. This

saves both cost and time when redesigning the board. This also makes the backplane significantly more adaptable, especially during the initial development phase when routing changes are expected.

**Table 2.** Comparison of CPLD family used in CubeSats.

| Satellite | CPLD Function | CPLD Family | Propagation Delay (Max) | Voltage Supply (Internal) |
|---|---|---|---|---|
| AAReST MirrorSat [18] | On-board Computer (OBC); Support for switching between the two Raspberry Pi compute modules and power sequencing | Xilinx XC9500 XL | 6 ns | 3~3.6 V |
| OPTOS [19] | Distributed On-board Data-Handling Terminal (DOT); Interface between the CPU and mission experiments | Xilinx Cool Runner II | 7 ns | 1.7~1.9 V |
| ARISSat-1 [20] | Integrated Housekeeping Unit (IHU); Glue logic between the video input processor, the SDRAM, and MCU | Intel Altera MAX II | 5.4 ns | 2.5 V, 3.3 V |
| BIRDS-3 [21] | Backplane; Interface between the main PIC and payloads | Lattice ispMach4000ZE | 5.8 ns | 1.7~1.9 V |

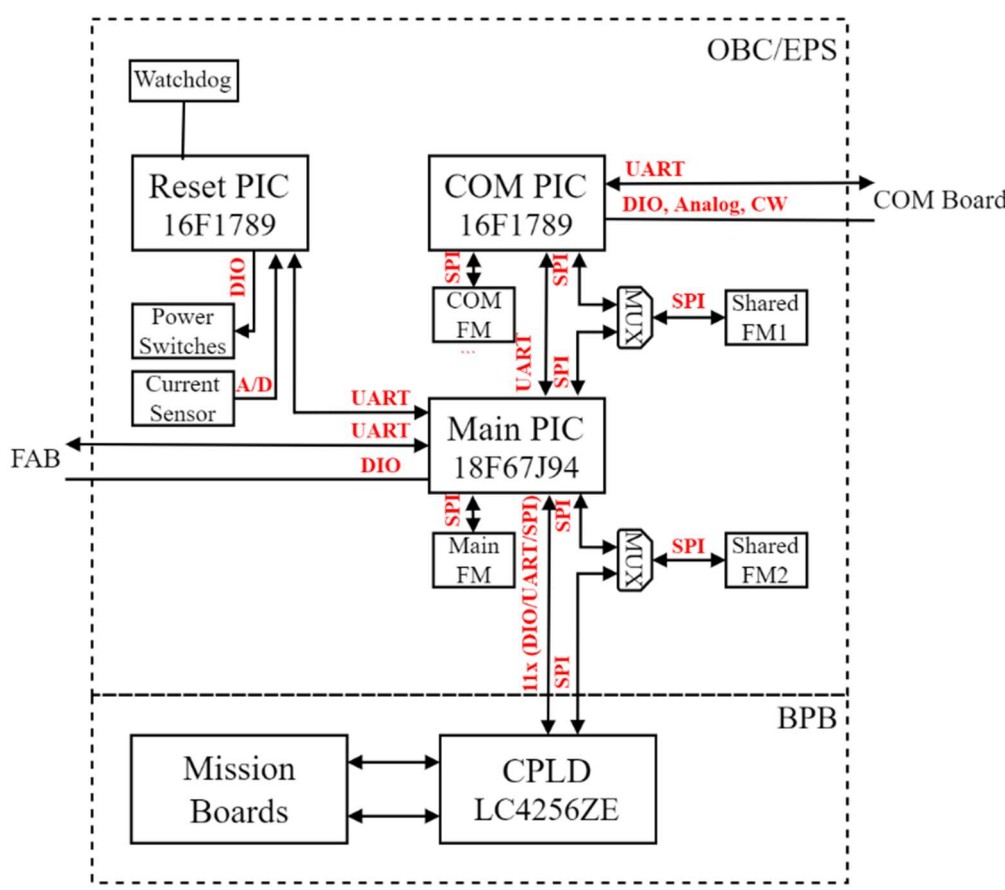

**Figure 4.** Block diagram of data handling between the bus and the payload.

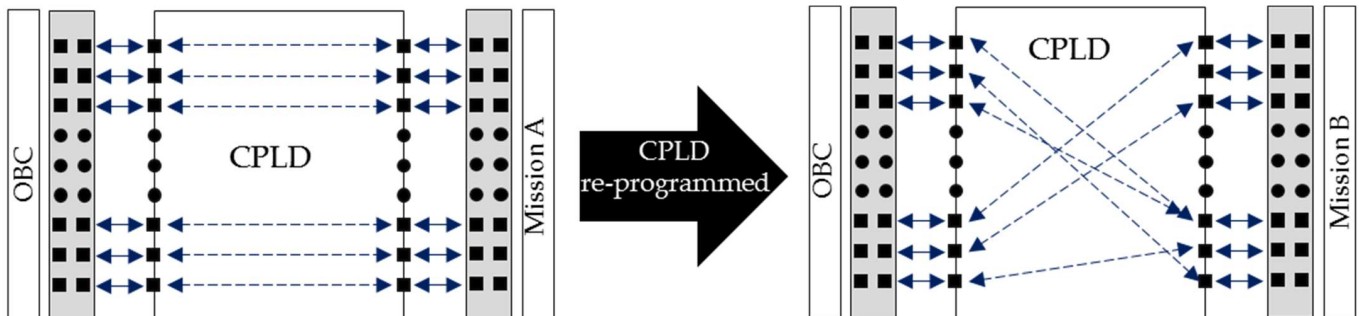

**Figure 5.** Digital line routing using a CPLD.

Using this standard bus system, the configurable interface board can be scaled up and adapted to different CubeSat sizes. As a result, resources can be more focused on developing the mission payload and instruments. Lastly, it allows more time for integration and system-level verification, which are critical for a reliable and successful mission [22,23]. Since implementing a configurable backplane to larger platforms has not been achieved before, challenges such as managing communication between the existing standard bus system and multiple mission payloads, as well as meeting the mission requirements of complex payloads, may occur. To prove that the standard bus system can be scaled and address the challenges encountered when scaling up to larger CubeSat platforms, two cases have been studied. In Section 2.1, a backplane prototype was developed to handle several missions based on the limited number of available electrical interfaces of a standard bus. In Section 2.2, an actual implementation of a backplane that can handle missions with complex requirements was demonstrated.

### 2.1. 3U-Size Configurable Backplane

The BIRDS bus system is not only intended for 1U CubeSats but is designed to scale up to a 3U platform with minor modifications [24]. One modification is in the design of the backplane, where a larger CubeSat provides more space for mission boards than in a 1U. A 3U configurable backplane prototype was developed, as shown in Figure 6. The backplane is a six-layer PCB and measures 320 mm × 90 mm × 1.6 mm, and all internal boards and deployment switch connectors are placed on the top side of the board, which is the same as the 1U standard bus. In addition, the bus system components (FAB, OBC/EPS, COM, and RAB) and their arrangement were kept unchanged. Lastly, the space between the COM board and the RAB was allocated for the 13 mission boards. Since the 3U platform would require more power for the mission payloads, the battery capacity would also increase, leading to a bigger battery box than in the 1U. Therefore, the space for the battery box, which is between the FAB and OBC/EPS board, is wider than in the 1U backplane.

A standard pin assignment for all mission boards is described in detail in Table 3. While 12 power pins are pre-assigned based on the power distribution from the OBC/EPS board, the developers have the flexibility to assign the remaining pins to CPLD, miscellaneous, and umbilical connections. In addition, the 20 pins assigned to CPLD connections are configurable even after the backplane is fabricated. This number of pins allows the payload developers to assign any kind of serial digital interface to link over the bus.

**Table 3.** Standard 50-pin assignment for mission boards.

| 50-Pin Mission Board Allocation | No. of Pins |
|---|---|
| CPLD connections | 20 |
| Miscellaneous (e.g., analog, direct connections) | 8 |
| Power | 12 |
| Umbilical (programming and debugging) | 10 |
| **Total** | **50** |

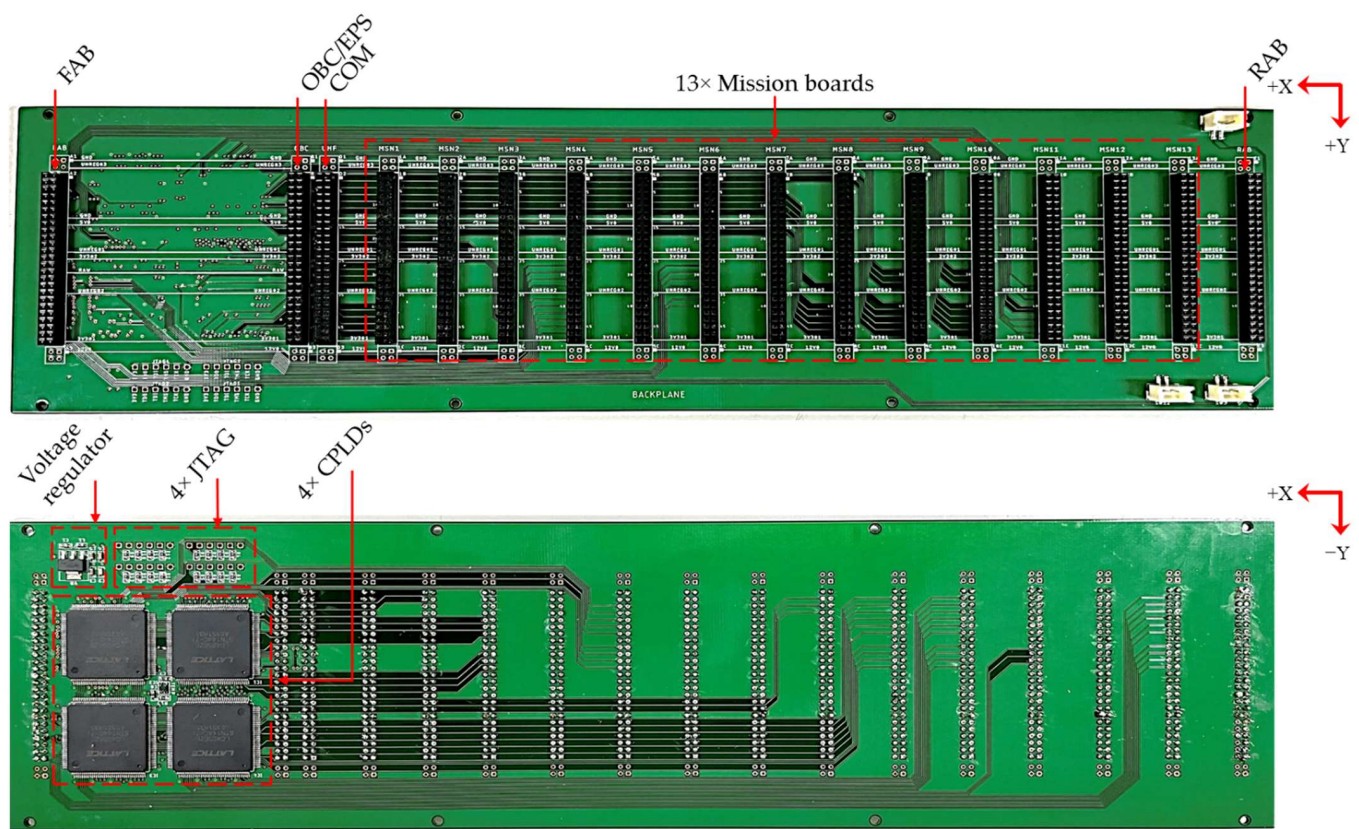

**Figure 6.** The 3U configurable backplane (**top** and **bottom** sides).

Four CPLDs are laid on the bottom side of the backplane, while the battery is placed on the other side. Cumulatively, the CPLDs provide the interface between the bus system and mission payloads. A voltage regulator supplies 1.8 V to all CPLDs, and each device has its JTAG pins for programming. Figure 7 shows the logical connections between the bus and the mission payloads. The SPI (from the shared FM) and 11 digital interfaces (from the main PIC) in the bus system are directly connected to CPLD1. The CPLDs are cascaded to each other through the 15 I/O pins. These are later configured to correspond to the bus system SPI and 11 digital interfaces. The remaining I/O pins of the CPLD are distributed to the mission (MSN) boards. CPLD1 to CPLD3 manage three mission boards, while CPLD4 manages four. To manage the bus system digital interfaces, the CPLDs are programmed to function as four-to-one multiplexers with four select (SEL) lines.

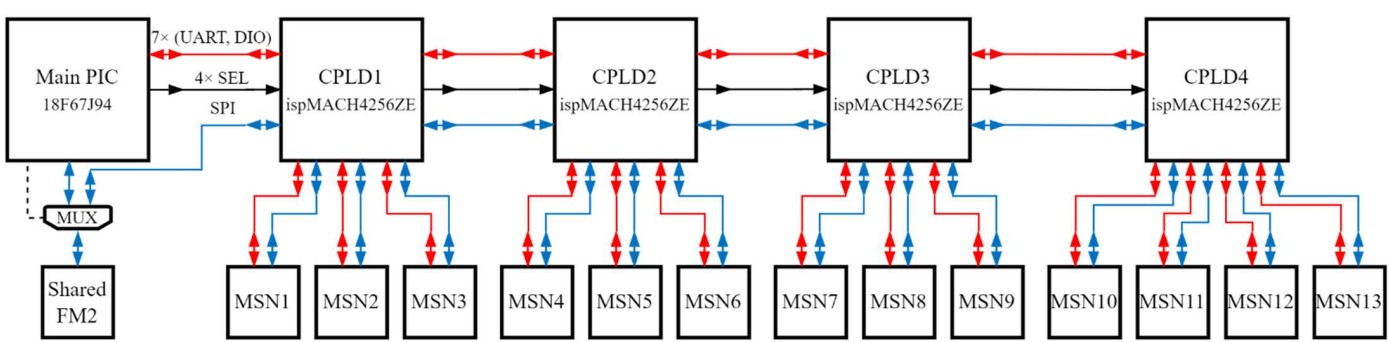

**Figure 7.** Logical connection of bus system and payload through CPLDs.

Table 4 shows the truth table of how the CPLDs function. A total of 4 of the 11 digital interfaces (SEL0 to SEL3) in the bus are used as select pins of the multiplexer function of the CPLDs. A specific logic state of the select pins allows a mission payload to access the

seven remaining digital interfaces and the SPI in the bus system. For example, if the select pin logic values are 0001, CPLD1 allows MSN1 to access the bus system. Conversely, if the select pin logic values are 0110, CPLD1 routes all 15 digital lines to CPLD2. CPLD2 then routes the digital lines to MSN6. There are 13 combinations of select pin logic states corresponding to the 13 mission boards.

**Table 4.** The 3U backplane truth table.

| Input | | | | Output | | | |
|---|---|---|---|---|---|---|---|
| SEL3 | SEL2 | SEL1 | SEL0 | CPLD1 | CPLD2 | CPLD3 | CPLD4 |
| 0 | 0 | 0 | 1 | MSN1 | - | - | - |
| 0 | 0 | 1 | 0 | MSN2 | - | - | - |
| 0 | 0 | 1 | 1 | MSN3 | - | - | - |
| 0 | 1 | 0 | 0 | CPLD2 | MSN4 | - | - |
| 0 | 1 | 0 | 1 | CPLD2 | MSN5 | - | - |
| 0 | 1 | 1 | 0 | CPLD2 | MSN6 | - | - |
| 0 | 1 | 1 | 1 | CPLD2 | CPLD3 | MSN7 | - |
| 1 | 0 | 0 | 0 | CPLD2 | CPLD3 | MSN8 | - |
| 1 | 0 | 0 | 1 | CPLD2 | CPLD3 | MSN9 | - |
| 1 | 0 | 1 | 0 | CPLD2 | CPLD3 | CPLD4 | MSN10 |
| 1 | 0 | 1 | 1 | CPLD2 | CPLD3 | CPLD4 | MSN11 |
| 1 | 1 | 0 | 0 | CPLD2 | CPLD3 | CPLD4 | MSN12 |
| 1 | 1 | 0 | 1 | CPLD2 | CPLD3 | CPLP4 | MSN13 |

Verifications were conducted on the backplane to test the performance. First, a functional test was performed to check whether the multiplexing function worked. Signal propagation delay and overall power consumption were also measured. Lastly, a bit error check was performed. The results and discussions of the test are given in Section 3.1.

### 2.2. KITSUNE W6U CubeSat

The previous section explained how a configurable backplane is designed to handle multiple payloads with relatively basic requirements that are related to power and data communication to the bus system. As a result, the bus system is not modified for integration. In addition, the payload operation does not require control of the satellite attitude. Therefore, the attitude determination and control subsystem (ADCS) was not included in the bus system. To implement payloads with advanced requirements, this section describes how a configurable backplane was modified, including additional subsystems.

KITSUNE satellite is a W6U CubeSat platform designed and developed in Japan as a collaboration project by the Kyushu Institute of Technology (Kitakyushu, Japan), Harada Seiki Co., Ltd. (Hamamatsu, Japan), and Addnics Corp. (Tokyo, Japan) [25]. The satellite project kicked off in September 2019, and KITSUNE was delivered to the Japan Aerospace Exploration Agency (JAXA) in November 2021. The satellite was deployed from the ISS on 24 March 2022 and is now in operation.

Figure 8 shows the 3D model of the W6U CubeSat. The CubeSat is divided into three sections: a 3U section for the camera payload that can capture 5 m class resolution images, a 1U section with technology demonstration and scientific experiment missions, and a 2U section for the main bus system. The 1U section is known as SPATIUM-II. It is basically a 1U satellite with its own bus that manages the missions. In addition to drawing power from the main bus, SPATIUM-II can work independently. A configurable backplane, developed to serve as the interface to all three sections, was placed in the middle of the satellite.

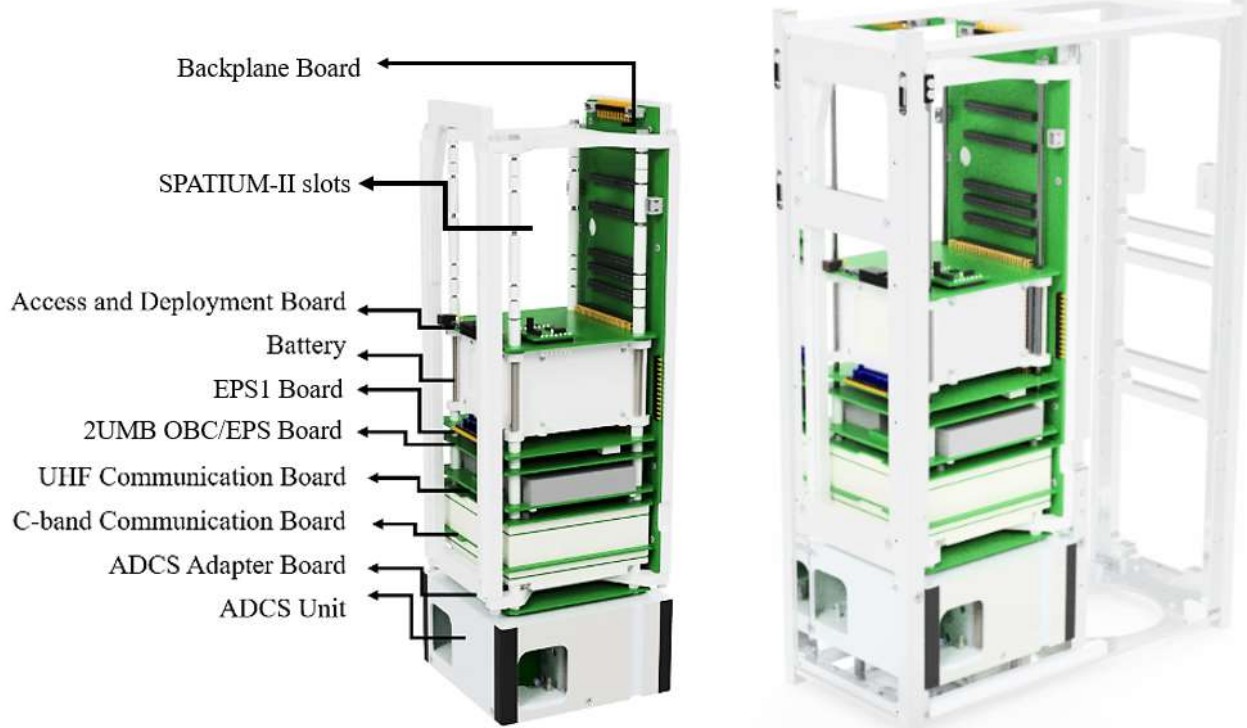

**Figure 8.** A 3D model of the W6U CubeSat bus system rack assembly (**left**) and the integrated rack assembly in the structure (**right**).

The main bus system was modified according to the mission requirement. The modifications were as follows:

- *FAB.* The functions of the FAB were separated into two boards. The umbilical to the bus system was assigned to the access and deployment board (ADB). The ADB provides external access to the microcontrollers (the main PIC, reset PIC, and COM PIC) for programming and debugging, battery charging, and voltage monitoring. The ADB has a microcontroller for antenna deployment. Power-related functions such as power generation, battery management, and power safety were assigned to the EPS1 board, as shown in Figure 8.

- *Battery.* The bus uses six commercial lithium-ion (Li-ion) batteries in a two-series, three-parallel (2S3P) configuration. The batteries are housed in a battery box and are connected to the FAB via a 12-pin connector.

- *OBC/EPS.* The three unused pins in the 50-pin connector were utilized—pin 3 was connected to a DIO of the main PIC, while pins 40 and 41 were connected to two DIOs of the COM PIC. The additional DIOs of the main PIC make the digital interfaces between the bus and payload available to 12 pins. Another radio transceiver can use the two DIOs of the COMP PIC. Lastly, the reset PIC provides an additional unregulated, 12 V power line for the camera payload [26].

- *Communication.* A C-band transceiver was utilized in addition to the UHF transceiver. The C-band transceiver is used for the high-speed download of data necessary in high-resolution images. It can also receive uplink commands as a backup to the UHF transceiver. The UHF (COM1) and C-band (COM2) transceivers use amateur radio bands.

- *ADCS.* The ADCS is necessary to obtain quality Earth images. A 0.5U ADCS from Adcole Maryland Aerospace was selected. This plug-and-play module has a standalone computer that manages its three reaction wheels, a three-axis magnetometer, two Earth

sensors, and three electromagnets [27]. An adapter board was used to connect the module to the backplane.

On the SPATIUM-II side, the components are as follows:

- *Access board (AB)*. This board is where the umbilical for SPATIUM-II was placed. It has external access to the OBC/EPS board and payload microcontrollers for programming and debugging. It is similar to the main bus ADB, except that it does not have a microcontroller for deployment.
- *OBC/EPS*. The three unused pins (pins 3, 40, and 41) in the 50-pin connector were connected to the DIO pins of the main PIC. The additional DIOs of the main PIC make the digital interfaces between the bus and the payload available to 14 pins. Lastly, the two 3.3 V lines distributed by the reset PIC were changed to 3.5 V and 4.5 V, as required by the payloads.
- *COM*. This is similar to the UHF transceiver in the BIRDS 1U bus system, except that it operates in the non-amateur UHF band.
- *Payloads*. SPATIUM-II has two missions, store-and-forward and total electron content (TEC). Store-and-forward uses an on-board LoRa payload that collects sensory data from ground sensor terminals (GST) and downlinks the data to the Kyutech non-amateur ground station. On the other hand, the TEC mission measures the total electron content in the ionosphere. The mission requires two connectors to the backplane for its payloads. The first payload is for the chip-scale atomic clock (CSAC) module [28]. The second payload has the Raspberry Pi module, software-defined radio (SDR), and radiofrequency (RF) switch. Lastly, the mission's global positioning system (GPS) receiver was placed in one of the solar panels.

The backplane board is a six-layer PCB with dimensions of 250.5 mm × 90 mm × 1.6 mm, as shown in Figure 9. The top side has connectors to the internal components of the main bus and the SPATIUM-II. All are 50-pin male connectors with a 2 mm pitch. The OBC/EPS, EPS1, and ADB in the main bus have additional 4-pin female connectors on each end of the 50-pin connector, which are allocated for the additional power lines, system ground, and battery power. The additional pins make the total pin count 58 for the three boards.

The two biggest solar panels (+X and −X) are connected to the backplane via 13-pin male connectors (A), while the −Y and +Z panels are connected via a 26-pin male connector (B). All connectors have a 2.54 mm pitch between pins. The fourth solar panel (+Y) is connected directly to the EPS1 board. In addition to routing the generated power and temperature readings to the EPS1 board, the connectors connect to the sun sensors and antenna deployment circuits. Additionally, B has a route to the two GPS modules. Lastly, C denotes two-pin male connectors for the deployment switches connected to the satellite structure.

At the bottom side of the backplane, there is a two-pin connector (D) that connects the sun sensor in the -Z panel and a 30-pin connector (E) that connects the camera controller to the backplane. There are two CPLDs for the main bus and SPATIUM-II working independently. A voltage regulator in the main bus converts 3.3 V from the power line to 1.8 V. There are two voltage regulators on the SPATIUM-II side. The first regulator converts 5 V from the power line to 3.3 V. Then, the second regulator converts 3.3 V to 1.8 V. The CPLD uses 3.3 V as the output supply voltage, whereas 1.8 V is the LVCMOS supply voltage.

Both CPLDs in the KITSUNE satellite function as voltage followers. The available interfaces on the bus are enough for the payloads to use. Thus, the multiplexing function is not necessary. The main bus CPLD has 43% utilization, while the SPATIUM-II CPLD has 35% utilization. This means that, out of the 96 I/O pins in a CPLD, the main bus and SPATIUM-II utilized 42 and 34 pins, respectively. These digital connections are combinations of DIO, SPI, and UART interfaces. Ground verification and on-orbit results are discussed in Section 3.2.

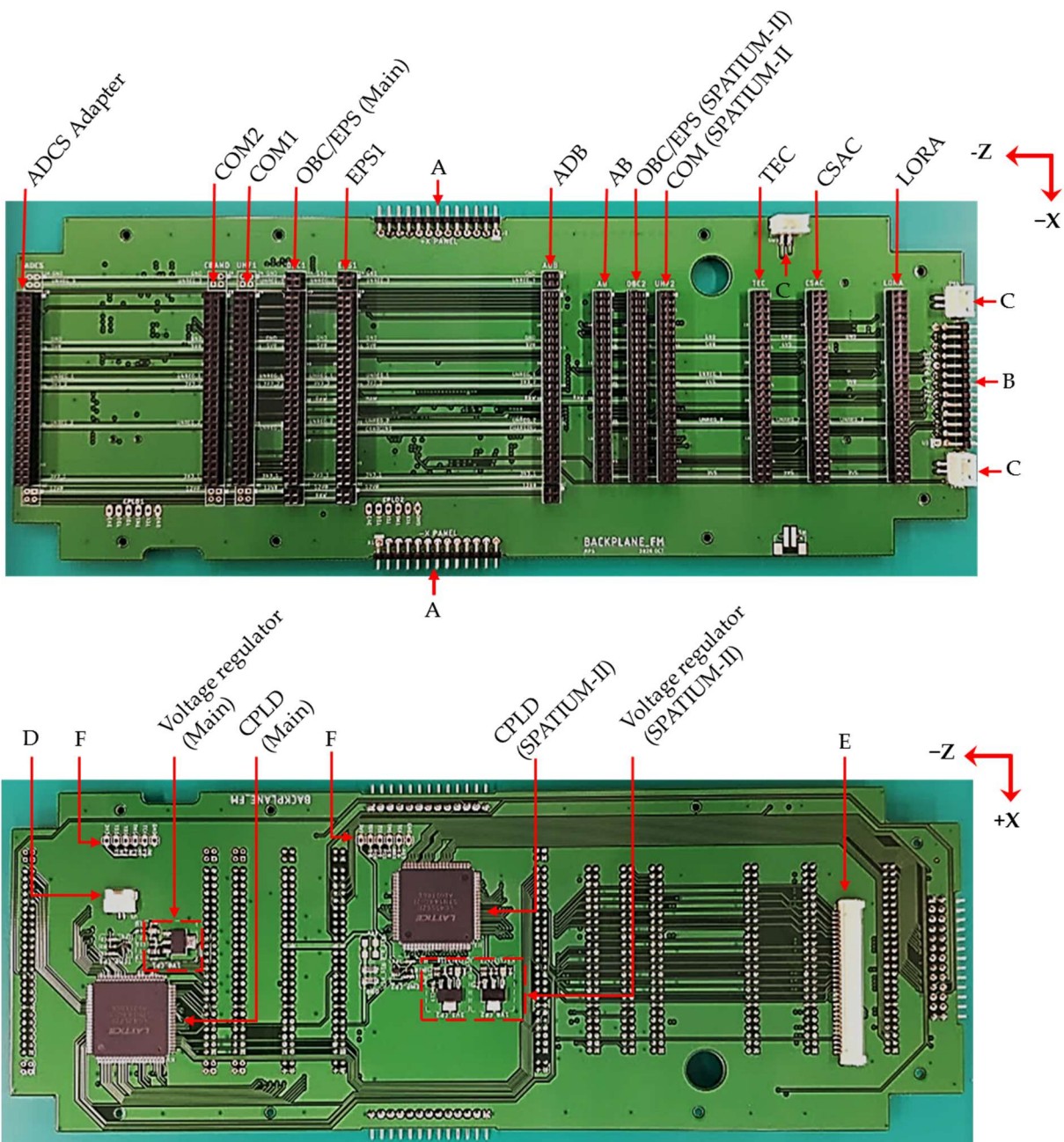

**Figure 9.** KITSUNE configurable backplane (**top** and **bottom** sides).

## 3. Tests and Results

This section discusses the tests that were conducted and the results. The first subsection covers the 3U backplane. The second subsection covers the KITSUNE backplane ground tests and on-orbit results.

### 3.1. 3U Backplane Verification

The four CPLDs in the backplane served as multiplexers, allowing the 13 mission boards to access the bus system's digital interfaces. Each CPLD's code was generated using very high-speed integrated circuit description language (VHDL). Lattice ispLEVER Classic was the design environment tool used to complete device design, including concept, synthesis, and simulation, as well as to generate the device joint electron device engineering council (JEDEC) programming file. Lastly, the JEDEC file was loaded into Lattice Diamond Programmer to program the CPLD via its JTAG pins.

The backplane board's functionality was validated by comparing the input and output signals of the OBC/EPS and mission board interfaces. To pass the functionality test, the two signal waveforms must be identical. The combination of logic levels on the four select pins determined which mission board had access to the bus interfaces. For example, in Figure 10, the select pins (9–12) were set to 0001. This combination allows Mission 1 access to the bus. The Digilent Digital Discovery instrument was used in the test as both a pattern generator and a logic analyzer. The instrument generated a 1 MHz clock as input to the OBC/EPS board, and the output signal from Mission 1 was compared to the input clock using the logic analyzer function. According to the waveforms, the output signal followed the logic values of the input signal. The same test was run on each of the 13 mission boards, and no differences were found.

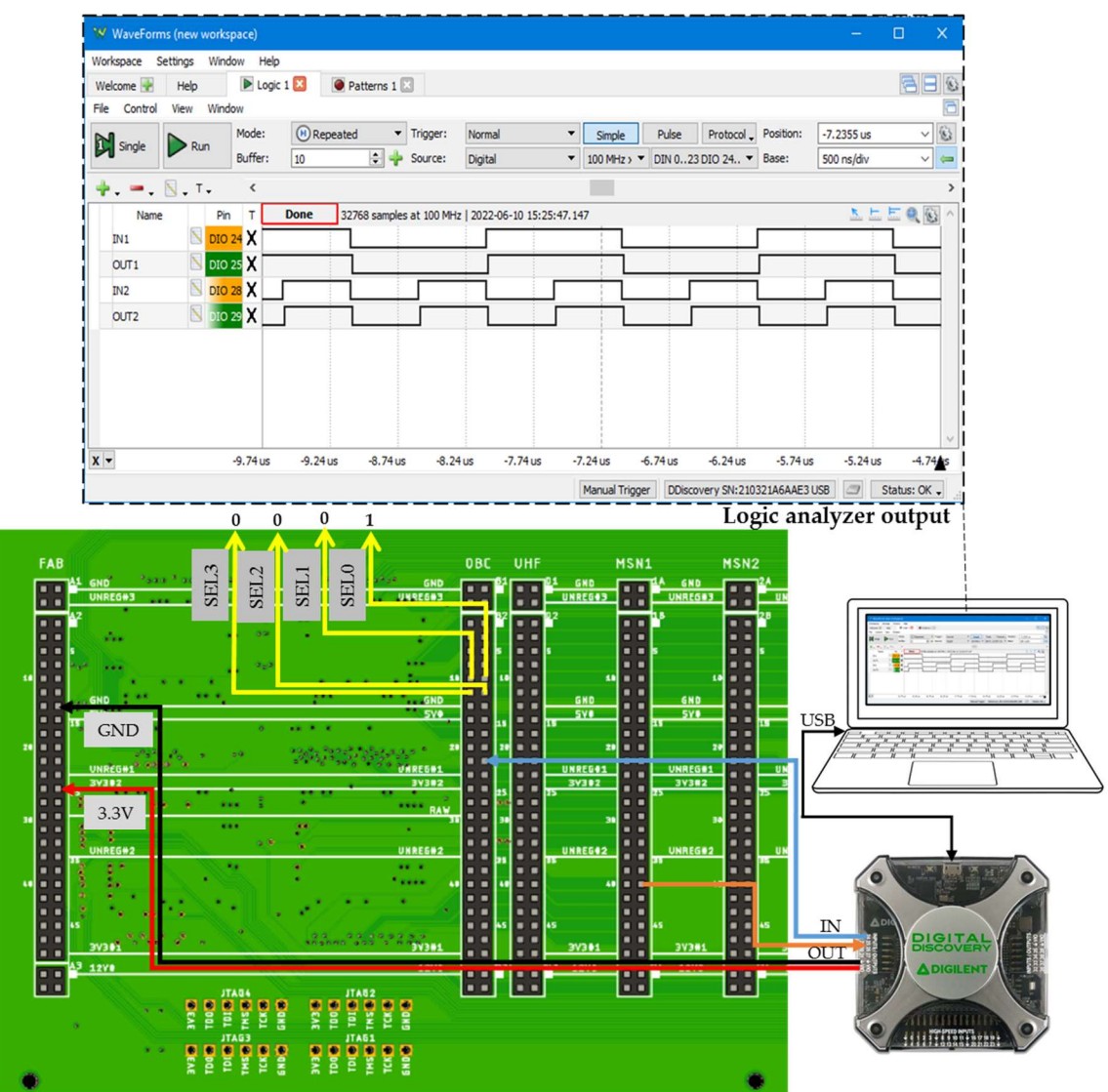

**Figure 10.** Functional test setup.

A bit error test was also performed to further validate the functionality of the backplane. For this test, a data stream was sent to the OBC/EPS digital interface and received by a mission through a CPLD. The data received from the mission were then compared to the data transmitted to check for possible bit differences. As illustrated in Figure 11, the Raspberry Pi module transmitted a bit array every 300 ms, and it reported the number of bit errors when it detected differences between transmitted and received data. This method was repeated for three different baud rates, such as 1 Mbps, 2 Mbps, and 4 Mbps, on each

mission board. While all mission boards had no bit differences recorded, the findings demonstrated the reliability and integrity of the backplane. One important note from this test is that the time difference between the transmission and reception of the data stream does not represent the transmission delay, which is explained next.

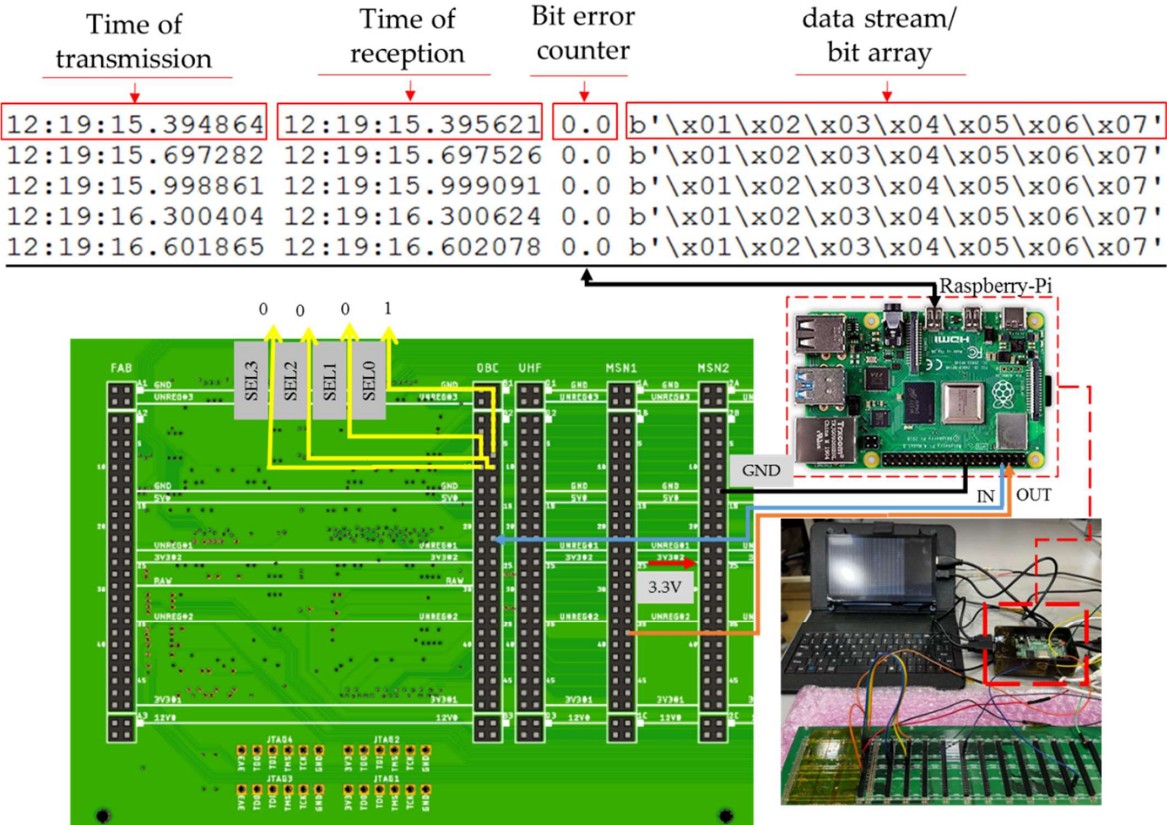

**Figure 11.** Bit error check test setup.

The transmission delay, defined as the time it takes for the data packet to arrive at the mission board, is another validated parameter. A significant delay may result in errors in the received data. Figure 12 shows the input and output signals as observed in the oscilloscope. The right photo shows that the two signals were identical. The waveforms were zoomed in for accurate delay measurements. All mission boards had their measurements taken, and Table 5 summarizes the measured transmission delay. The measured values were grouped based on the number of CPLDs through which the signal passed. In addition, the delay was measured twice for each mission board using different signal input sources—a Digilent Analog Discovery pattern generator (Test 1) and a function generator (Test 2). According to the results, the transmission delay of a single CPLD was approximately 5.0 ns. When a signal was transmitted through all four CPLDs, the transmission delay was measured as approximately 20.0 ns. Lastly, it is concluded that the measured transmission delay could not produce bit errors in data arrays.

Lastly, the power consumption of the backplane board was investigated since the available power for a small satellite platform such as CubeSats with limited resources determines survivability in orbit as well as the ability to support multiple payloads. The current consumption was measured at the 3.3 V input to the voltage regulator under two conditions as idle mode and active CPLDs. When all four CPLDs were active, the measured current increased from 4.3 mA to 5.6 mA. As a result, the maximum power consumption was determined as approximately 18.5 mW. This result confirmed two important points. First, the power drawn by the backplane would have negligible impact on the overall power consumption of a satellite. For instance, the backplane would only need 28.0 mWh of

energy per cycle in ISS orbit. Second, the four CPLDs consumed significantly low power compared to the voltage regulator. According to datasheets, the voltage regulator quiescent current was 4.0 mA, whereas the CPLD quiescent current was only 13 uA. Therefore, if it is necessary to reduce the power consumption of the backplane even further, the focus should be on the voltage regulator rather than the CPLD.

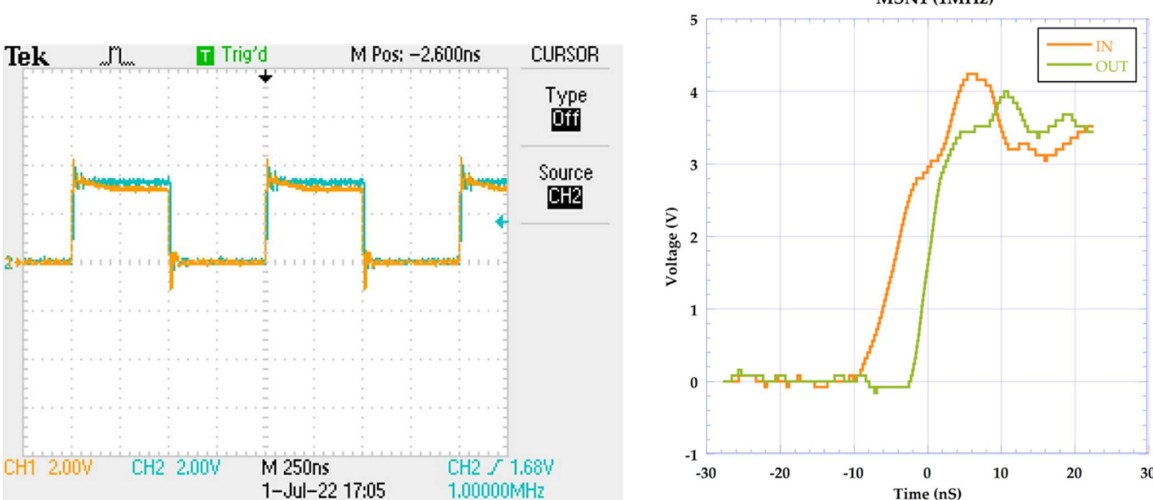

**Figure 12.** Input and output signals on a CPLD.

**Table 5.** Summary of the measured transmission delay.

| Mission Board | No. of CPLDs | Transmission Delay | |
|---|---|---|---|
| | | **Test 1** | **Test 2** |
| Mission 1–3 | 1 | 5.2 ns | 6.0 ns |
| Mission 4–6 | 2 | 9.4 ns | 10.2 ns |
| Mission 7–9 | 3 | 13.3 ns | 14.4 ns |
| Mission 10–13 | 4 | 18.5 ns | 16.8 ns |

*3.2. KITSUNE Backplane Verification*

During the satellite's development, board- and system-level verifications were performed. In addition, satellite on-orbit data were available. The subsections that follow discuss both ground and on-orbit results.

3.2.1. Ground Tests

The main bus, as well as the SPATIUM-II CPLDs in the backplane, served as voltage followers. The input and output signals were compared during the board-level verification. The same test method as was used on the 3U backplane prototype was used. Table 6 lists the digital interfaces that were routed to the CPLD. The main bus CPLD routed four pairs of UART, two sets of SPI, and five DIO lines. The SPATIUM-II CPLD routed three pairs of UART, one SPI, and seven DIO lines. The data confirmed that the CPLD could route serial interfaces such as UART and SPI.

**Table 6.** Summary of digital interfaces routed through CPLD.

| Interfaces | No. of CPLD Pins | |
|---|---|---|
| | **Main Bus** | **SPATIUM-II** |
| UART | 16 | 12 |
| SPI | 16 | 8 |
| DIO | 10 | 14 |
| **Total** | **42** | **34** |

A functional test to check for bit errors was performed in the KITSUNE backplane, which is similar to the 3U backplane prototype. In the test, the Raspberry Pi module transmitted a data stream to the OBC/EPS every 300 ms, and the device then compared the signal received from the ADCS adapter board. When a difference between the two datasets was detected, the Raspberry Pi module displayed the number of bit errors. The test was conducted at temperatures ranging from −10 °C to +70 °C in a vacuum chamber. There was no recorded bit error during the entire 12 h test.

Two backplane-related failures were observed during system integration. During the engineering model (EM) development, a line connecting an umbilical in the ADB to the EPS1 was accidentally routed through the main bus CPLD. It was found that the line was used to monitor the battery voltage. The CPLD was severely damaged due to the error, so the ICD was thoroughly reviewed before finalizing the design to prevent the incident from reoccurring. Another recorded failure was in one of the main bus SPI interfaces. The ADCS adapter board's microcontroller could not obtain data from the ADB's magnetometer placed on the ADB, and it was found that the memory input slave output (MISO) line's signal direction was incorrect. This issue was quickly resolved by updating the VHDL code and reprogramming the CPLD.

The operation and performance of the flight model (FM) satellite were tested in various space environment conditions to demonstrate that the satellite could operate properly in low Earth orbit. A summary of the thermal vacuum and vibration tests is shown in Table 7.

**Table 7.** Space environment test parameters.

| Test | Information |
| :---: | :---: |
| **Thermal Vacuum Test** | |
| Temperature range | −15 °C to +60 °C |
| Number of cycles | 2 |
| **Vibration Test** | |
| Acceleration level | 5.77 Grms |
| Duration | 2 min |

The telemetry reading from the thermal vacuum test (TVT) in Figure 13 shows that the main bus CPLD temperature was about 2 °C higher than that of the SPATIUM-II CPLD. This is because the main bus CPLD was placed right below the battery, which is a heat source, for the entire test duration. The TVT ran for two cycles, and the recorded minimum and maximum CPLD temperatures were −3 °C and +50 °C. There was no recorded anomaly in the performance of the backplane for the entire 87 h duration.

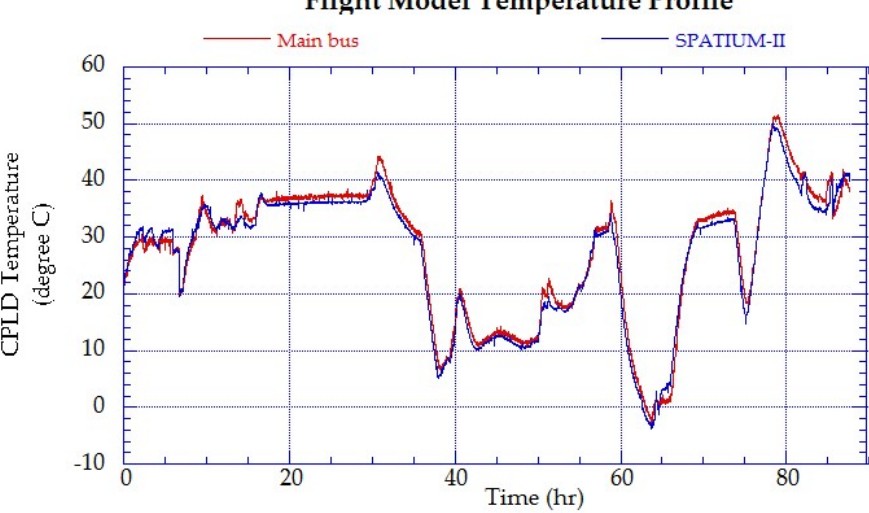

**Figure 13.** TVT CPLD temperature profile.

### 3.2.2. On-Orbit Results

Since the deployment of KITSUNE in March 2022, data communications between components in the main bus have been confirmed. For example, a ground command to activate the ADCS and set a specific ADCS mode was sent to the satellite. When the command was received, the main PIC sent a trigger through a DIO interface to activate the overcurrent protection (OCP) circuit that powered up the ADCS PIC. The main PIC then used UART lines to communicate with the ADCS PIC. The satellite log confirmed this set of actions. When a ground command to download ADCS telemetry data was sent to the satellite, the data stored in the ADCS FM were transferred to the bus system's shared FM. The data from the shared FM were then accessed by the COM PIC (via SPI) and downloaded to the ground. The downloaded satellite log and ADCS telemetry data confirmed that the routes between the bus and the ADCS were operational. DIO, UART, and SPI interfaces were connected to the CPLD. As a result, the CPLD was carrying out its function. Table 8 summarizes the on-orbit data communications via the main bus CPLD; the circles (○) in the last column indicate that all 21 digital lines were verified to be working.

**Table 8.** Summary of on-orbit data communications in the main bus through CPLD.

| Digital Lines | Baud Rate | On-Orbit Result |
|---|---|---|
| 2× UART (Main PIC–ADB PIC) | 9600 | ○ |
| 2× UART (Main PIC–ADCS PIC) | 9600 | ○ |
| 2× UART (Main PIC–CBAND) | 115,200 | ○ |
| 2× UART (COM PIC–CBAND) | 115,200 | ○ |
| 4× SPI (ADCS PIC–Magnetometer) | 1,000,000 | ○ |
| 4× SPI (ADCS FM–Shared FM) | 1,000,000 | ○ |
| DIO (Main PIC–ADB OCP) | - | ○ |
| DIO (ADCS PIC–Magnetometer reset) | - | ○ |
| DIO (ADCS PIC–Magnetometer DRDY) | - | ○ |
| DIO (Main PIC–ADCS OCP) | - | ○ |
| DIO (COM PIC to CBAND CW) | - | ○ |

The same verification was performed in the SPATIUM-II section. For example, a ground command to activate the LoRa payload was sent to the satellite. When the command was received, the main PIC sent a trigger through a DIO interface to activate the overcurrent protection (OCP) circuit that powered up the LoRa MCU. The main PIC then used UART lines to communicate with the LoRa MCU. The data from the LoRa payload were directly stored in the bus system's shared flash memory. The data from the shared flash memory were then accessed by the COM PIC (via SPI) and downloaded to the ground. The downloaded satellite log and LoRa data confirmed that the routes between the bus and the LoRa payload were operational. DIO, UART, and SPI interfaces were connected to the CPLD. As a result, the CPLD was carrying out its function. Table 9 summarizes the on-orbit data communications via the main bus CPLD. The circles in the last column indicate that all 15 digital lines were verified to be working.

**Table 9.** Summary of on-orbit data communications in the SPATIUM-II bus through CPLD.

| Digital Lines | Baud Rate | On-Orbit Result |
|---|---|---|
| 2× UART (Main PIC–LoRa MCU) | 19,200 | ○ |
| 2× UART (Main PIC–RPI) | 19,200 | ○ |
| 2× UART (Main PIC–CSAC) | 19,200 | ○ |
| 4× SPI (LoRa /MCU–Shared FM) | 1,000,000 | ○ |
| 3× DIO (Main PIC–BC OCP) | - | ○ |
| DIO (Main PIC–LoRa OCP) | - | ○ |
| DIO (Main PIC–RPI OCP) | - | ○ |
| DIO (Main PIC–GPS) | - | ○ |
| DIO (Main PIC to SDR) | - | ○ |

Figure 14 shows that the on-orbit power consumption of the main bus and the SPATIUM-II was comparable to the ground data. The average power consumption per orbit of SPATIUM-II was 65 mW, while that of the main bus was 16 mW. The number of voltage regulators explains the difference in power consumption between the main bus and SPATIUM-II. The main bus has a 1.8 V regulator, whereas SPATIUM-II has 3.3 V and 1.8 V regulators. We can recall that CPLD uses 1.8 V as the supply for its LVCMOS and 3.3 V as the supply for the output logic voltage. Since SPATIUM-II does not have a 3.3 V power line, it uses the 5 V power line and first converts it to 3.3 V, then 1.8 V. The 1.8 V and 3.3 V regulators have standby currents of 4 mA and 8 mA, respectively.

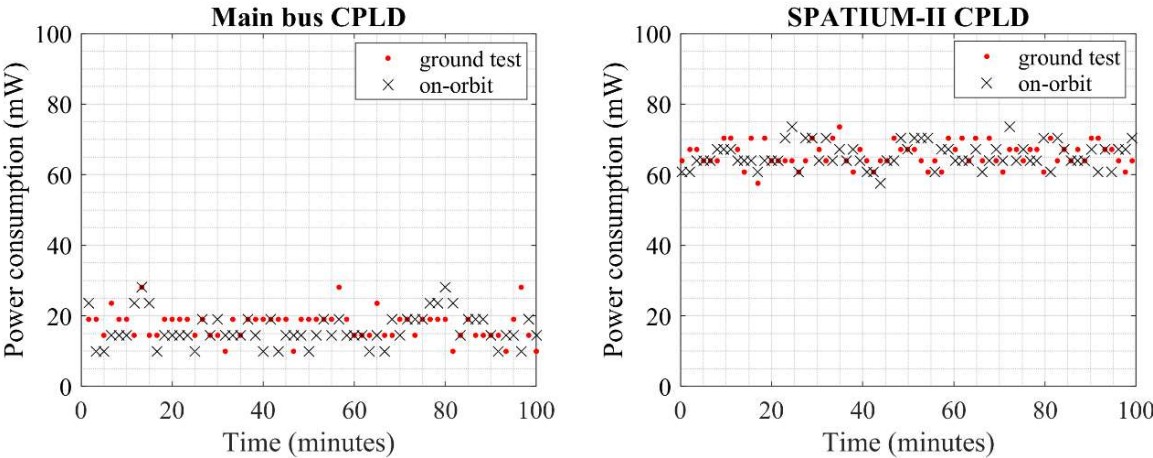

**Figure 14.** Ground data vs. on-orbit data power consumption for 2U main bus (**left**) and SPATIUM-II (**right**).

In Figure 15, four temperature profiles of the CPLDs per one orbit in different Sun beta angles were plotted. All graphs show that the main bus CPLD (CPLD1) temperature was at least 3 °C higher than that of the SPATIUM-II CPLD. This observation was the same as the TVT result, where the temperature difference was attributed to the main bus CPLD being placed close to the battery. The direct relation of the Sun beta angle to the device temperature was also evident in all four graphs.

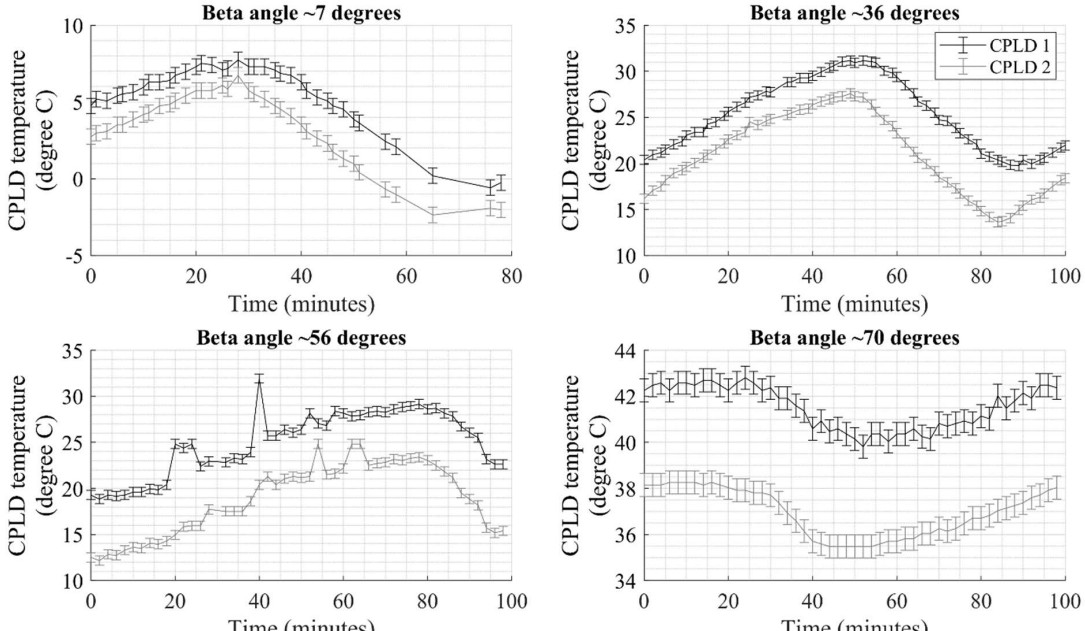

**Figure 15.** On-orbit CPLD temperature profile at different Sun beta angles.

*3.3. Summary of Tests and Results*

From the two cases above, the configurability of the backplane was demonstrated, and the following parameters were confirmed:

- *Number of configurable pins.* A maximum of 20 out of 50 pins on each mission board were made configurable. This number of pins allows the payload developers to assign any kind of serial digital interface to link over the bus.
- *Transmission bit errors.* The integrity of the data received over the configurable backplane was confirmed in two different instances. First, data were transmitted at baud rates up to 4 Mbps at room temperature. Second, the test was conducted in a vacuum environment at temperatures ranging from $-10\,°C$ to $+70\,°C$ for 12 h at a 1 Mbps baud rate. No bit errors were detected during either test, and the findings demonstrated the reliability and integrity of the backplane.
- *Transmission delay.* A maximum transmission delay of 20 ns over four CPLDs in the backplane was observed, which is acceptable for simple serial communications such as UART and SPI.
- *Power consumption.* The maximum power consumed by the backplane with four CPLDs was 18.5 mW or 28.0 mWh of energy per cycle in ISS orbit. The power drawn by the backplane is insignificant to the overall power consumption of a satellite.
- *Development time.* From the experience of satellite development, a simple routing error in a hardwired backplane would take at least three weeks and additional costs to fix. However, in the case of a configurable backplane, a simple change in the CPLD code could resolve the error within hours without incurring costs.

## 4. Discussions

The present paper studied the scalability of configurable electrical interfaces for two cases. In the first case, a 3U-size configurable backplane prototype was designed to support 13 mission payloads. The backplane contained four CPLDs that served as multiplexers, allowing the bus system to control all payloads with a limited number of digital interfaces. Several tests were conducted to verify its functionality and performance. The results showed that the backplane would only consume 28 mWh of energy per cycle in ISS orbit. This is considerably low and does not affect the overall power consumption of a CubeSat. A total transmission delay of up to 20 ns was measured, which is acceptable for serial communications such as UART and SPI. The short transmission delay also ensured data integrity after it was transmitted through the CPLDs. This was validated further in the bit error check, where the transmitted signal (of up to 4 Mbps) was compared to the received signal, and no bit errors were detected throughout the test.

In the second case, a configurable backplane was used in a W6U satellite that carried out complex missions. The electrical bus system was modified as needed to meet the mission requirements. Additional power lines, integration of commercial ADCS and high-speed transceivers, and an autonomous bus system for the 1U SPATIUM-II section were among the modifications. According to the on-orbit results, no anomalies were detected on any of the 21 digital connections in the main bus or on any of the 17 digital connections in SPATIUM-II, which passed through the CPLDs. Furthermore, the power consumption of the satellite's main bus and SPATIUM-II sections in orbit was comparable to the ground results.

The advantage of this study is that the scalable standard bus allows more time for integration and system-level verification, which is critical for a reliable and successful mission. The design concept could also benefit satellite developers who provide hosted payload services where bus resources are maximized to accommodate as many payloads as possible. UART and SPI communications were extensively used in the study since the bus system's command and data handling are based on these protocols. However, other protocols have not been supported by the configurable backplane. According to Cho et al. [2], other protocols, such as I2C, CAN, USB, and Ethernet, can be used and are

expected to be used by CubeSat developers and vendors. A programmable backplane that supports these protocols should be developed in future work.

Turmenjargal et al. [14] recommended that the backplane is reconfigurable after the satellite has been fully assembled or even after the satellite has been launched into orbit. The former is easily implemented by inserting the JTAG pins into the umbilical. The 3U backplane prototype partially meets the latter recommendation. The preconfigured multiplexing function of the CPLDs in the backplane allows the selected payload(s) access to the bus system by sending ground commands to the satellite. Full on-orbit reconfiguration of the backplane CPLD as a contingency is still under investigation.

Kim et al. [24] stated that the flexibility of the BIRDS 1U electrical bus system is one of its key features. The BIRDS bus system was designed to be compatible with up to 3U platforms, with minimum modifications. The BIRDS bus command and data-handling architecture were retained in the 3U backplane prototype and the KITSUNE backplane. UART and SPI are the interfaces between the bus system and the payload. However, other protocols may be required by mission payloads. A bridge circuit capable of translating different protocols between the bus and payload is needed to keep the existing architecture. The bridge circuit can be implemented in the configurable backplane.

## 5. Conclusions

A backplane board provides an electrical interface among CubeSat components. It has additional advantages, such as ease of assembly/disassembly and fewer harnesses than a de facto standard PC/104 style interface. To confer more flexibility to the interface, a software-configurable backplane was developed. The backplane with an ispMACH LC4256ZE CPLD was demonstrated onboard a 1U CubeSat for more than 2 years in orbit. Through a CPLD, digital interfaces can be rerouted without requiring hardware changes by reprogramming the CPLD. This saves both money and time when redesigning the board.

The designed configurable interface board was verified to be scalable and adaptable to different CubeSat sizes while absorbing the challenges in the process. While a hardwired backplane is applied to a specific satellite, the scalable standard bus in this study exhibited its reusability in multiple satellite projects. Hence, it has the advantage of providing additional time for system-level integration and verification, which are essential for a reliable and successful mission. The design concept could also be advantageous to satellite manufacturers that offer hosted payload services, where bus resources are utilized to support many payloads.

Several future studies have been identified to fully realize the configurable backplane's scalability. The first is to verify that it can support communication protocols other than UART and SPI. Protocols such as I2C, CAN, and Ethernet are currently being used and are desired to be utilized in future projects by CubeSat developers and vendors. Secondly, a bridge circuit that translates different protocols can be incorporated into the configurable backplane. The bridge circuit will allow the use of the BIRDS electrical bus architecture on missions that require protocols other than UART and SPI. Lastly, on-orbit reconfiguration of the CPLD as a contingency could be explored.

**Author Contributions:** Conceptualization, M.S.; methodology, M.S.; software, M.S. and Y.O.; validation, M.S., Y.O., and N.C.O.; formal analysis, M.S.; investigation, M.S. and N.C.O.; resources, M.C.; data curation, M.S.; writing—original draft preparation, M.S.; writing—review and editing, N.C.O., T.Y., Y.O., and M.C.; visualization, M.S. and N.C.O.; supervision, M.C.; project administration, M.C.; funding acquisition, M.C. All authors have read and agreed to the published version of the manuscript.

**Funding:** This research was partially funded by JSPS Core-to-Core Program B: Asia–Africa Science Platforms (JPJSC2020005) and the "Acquisition and dissemination promotion project consignment fee for international standards related to energy conservation" by the Ministry of Economy, Trade and Industry, Japan.

**Institutional Review Board Statement:** Not applicable.

**Informed Consent Statement:** Not applicable.

**Data Availability Statement:** Not applicable.

**Acknowledgments:** The authors would like to express gratitude to the KITSUNE team members, particularly Jose Rodrigo Cordova Alarcon, Victor Hugo Schulz, Pooja Lepcha, Tharindu Lakmal Dayarathna Malmadayalage, Abhas Maskey, Adolfo Javier Jara Cespedes, Anibal Antonio Mendoza Ruiz, Cosmas Kiruki, Daisuke Nakayama, Dmytro Faizullin, Dulani Chamika Withanage, Ei Phyu Phyu, Eyoas Ergetu Areda, Fatima Gabriela Duran Dominguez, Hari Ram Shrestha, Hoda Awny A. A. Elmegharbel, Ibukun Oluwatobi Adebolu, Kateryna Aheiev, Kentaro Kitamura, Makiko Kishimoto, Mariko Teramoto, Mark Angelo Cabrera Purio, Masui Hirokazu, Mazaru Ariel Manabe Safi, Muhammad Hasif Bin Azami, Ofosu Joseph Ampadu, Sangkyun Kim, Takashi Oshiro, Victor Mukungunugwa, Yuma Nozaki, and Yuta Kakimoto. Their contributions to the development and operation of the satellite are highly appreciated.

**Conflicts of Interest:** The authors declare no conflict of interest.

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
