# Peer review of "Scalable and Configurable Electrical Interface Board for Bus System Development of Different CubeSat Platforms"

_applsci, doi:10.3390/app12188964_

Round 1

Reviewer 1 Report

The authors designed a flight-proven electrical bus system for 1U CubeSat platform in the BIRDS satellite program at Kyushu Institute of Technology. A backplane board serves as the mechanical and electrical interface between the bus subsystems and the payloads. The results look encouraging and motivating. But there are still some contents, which need be revised in order to meet the requirements of publish. A number of concerns listed as follows:

(1)   The abstract does not provide significant information and it should be revised to highlight the significant methodological contributions and conclusions.

(2)   In the Section 1, the main contributions of this paper should be further summarized and clearly demonstrated. This reviewer suggests the authors exactly mention what is new compared with existing approaches and why the proposed approach is needed to be used instead of the existing methods.

(3)   The method in the context of the proposed work should be written in detail.

(4)   All abbreviations need to be written in full for the first time, such as Lines 190-192, OBC/EPS, COM, and so on.

(5)   In the Section 2, the core idea and key steps should be provided in order to clearly describe the main work of the study.

(6)   Those related works and their relevance are required to analyze. You must add and review all significant similar works that have been done. For example, 10.3390/agriculture12060793; 10.1109/JSTARS.2021.3059451 ;10.1016/j.engappai.2022.105139ï¼› 10.1007/s10489-022-03719-6 and so on.

(7)   Result and discussion should be rewritten to summarize the significance of the work.

(8)   What are the advantages and disadvantages of this study compared to the existing studies in this area?

(9)   Correct typological mistakes and mathematical errors.

Author Response

Thank you very much for the comments and suggestions. The detailed response for each comment can be seen on the attached file.

Reviewer 2 Report

Authors have presented work on " Scalable and Configurable Electrical Interface board for bus system development of different CubeSat platforms". It is a good work but need minor revision prior to accept in journal.

1. Key findings of the work need to be addressed in the abstract. I suggest to rewrite the abstract.

2.  For configurable backplane implementation, what are the parameters you have taken? And compare it with proposed design.

3.  Can you check your design with another CPLD family and compare.

4. Add some recent literatures of your study.

Author Response

(The authors gave the same response as above.)

Round 2

Reviewer 1 Report

According to the revised paper, I have appreciated the deep revision of the contents and the present form of this manuscript. There is little content, which need be revised according to the comment of reviewer in order to meet the requirements of publish. A number of concerns listed as follows:

(1) Please highlight your contributions in introduction.

(2) How to determine these parameters? The author should give a detailed explanation.

(3) Conclusion: What are the advantages and disadvantages of this study compared to the existing studies in this area?

(4) Result and discussion should be rewritten to summarize the significance of the work.

(5)  In order to further highlight the introduction, some latest references in the previous should be added to the paper for improving the reviews part.

(6)  There are some grammatical mistakes and typo errors. please proof read from native speaker.

Author Response

On behalf of the co-authors, I would like to thank you for the provided comments and suggestions for the improvement of the paper. I would like to submit the reply to reviewer's comments for your perusal. 

For your kind consideration. Thank you very much.

Round 3

Reviewer 1 Report

I think that this paper can be accepted.